# Kribbellichelins A and B, Two New Antibiotics from *Kribbella* sp. CA-293567 with Activity against Several Human Pathogens

**DOI:** 10.3390/molecules27196355

**Published:** 2022-09-26

**Authors:** Jorge R. Virués-Segovia, Fernando Reyes, Sandra Ruíz, Jesús Martín, Ignacio Fernández-Pastor, Carlos Justicia, Mercedes de la Cruz, Caridad Díaz, Thomas A. Mackenzie, Olga Genilloud, Ignacio González, José R. Tormo

**Affiliations:** Fundación MEDINA, Av. Conocimiento 34, PTS Health Sciences Technology Park, 18016 Granada, Spain

**Keywords:** actinobacteria, *Kribbella*, structural elucidation, Marfey’s analysis, antifungal activity, antibacterial activity, cytotoxic activity, antibiotic activity

## Abstract

Current needs in finding new antibiotics against emerging multidrug-resistant superbugs are pushing the scientific community into coming back to Nature for the discovery of novel active structures. Recently, a survey of halophilic actinomyectes from saline substrates of *El Saladar del Margen*, in the Cúllar-Baza depression (Granada, Spain), led us to the isolation and identification of 108 strains from the rhizosphere of the endemic plant *Limonium majus*. Evaluation of the potential of these strains to produce new anti-infective agents against superbug pathogens was performed through fermentation in 10 different culture media using an OSMAC approach and assessment of the antibacterial and antifungal properties of their acetone extracts. The study allowed the isolation of two novel antibiotic compounds, kribbellichelin A (**1**) and B (**2**), along with the known metabolites sandramycin (**3**), coproporphyrin III (**4**), and kribelloside C (**5**) from a bioassay-guided fractionation of scaled-up active extracts of the *Kribbella* sp. CA-293567 strain. The structures of the new molecules were elucidated by ESI-qTOF-MS/MS, 1D and 2D NMR, and Marfey’s analysis for the determination of the absolute configuration of their amino acid residues. Compounds **1–3** and **5** were assayed against a panel of relevant antibiotic-resistant pathogenic strains and evaluated for cytotoxicity versus the human hepatoma cell line HepG2 (ATCC HB-8065). Kribbellichelins A (**1**) and B (**2**) showed antimicrobial activity versus *Candida albicans* ATCC-64124, weak potency against *Acinetobacter baumannii* MB-5973 and *Pseudomonas aeruginosa* MB-5919, and an atypical dose-dependent concentration profile against *Aspergillus fumigatus* ATCC-46645. Sandramycin (**3**) confirmed previously reported excellent growth inhibition activity against MRSA MB-5393 but also presented clear antifungal activity against *C. albicans* ATCC-64124 and *A. fumigatus* ATCC-46645 associated with lower cytotoxicity observed in HepG2, whereas Kribelloside C (**5**) displayed high antifungal activity only against *A. fumigatus* ATCC-46645. Herein, we describe the processes followed for the isolation, structure elucidation, and potency evaluation of these two new active compounds against a panel of human pathogens as well as, for the first time, the characterization of the antifungal activities of sandramycin (**3**).

## 1. Introduction

For decades, microbial secondary metabolites have played a key role in the development of new pharmaceutical agents due to their chemical diversity, structural complexity, and biological activities [1].

However, the recent emergence of antibiotic-resistant microbes has become a global health challenge for the scientific community. In fact, in 2019, about 4.95 million deaths were associated with bacterial antimicrobial resistance, including 1.27 million deaths attributable to this health problem [2].

In addition, traditional chemical investigation methods of bioactive natural products are becoming inefficient due to the frequent rediscovery of known entities or low production yields obtained under laboratory conditions [3]. In this sense, one of the most extended and straightforward approaches for exploiting the chemical diversity produced by microorganisms is the variation of the culture conditions (medium composition, aeration, temperature, pH, salinity, addition of resins, DMSO or EtOH), also known as the “one strain many compounds” (OSMAC) approach [4].

Attending to its biological activity, the Domain Bacteria has proven to be a major source of these structures, exhibiting some kind of bioactivity (47%), where at least 67% present antibiotic properties [5]. In this sense, actinobacteria are characterized by a rich metabolism able to produce a plethora of bioactive secondary metabolites due to their richness in still untapped biosynthetic gene clusters (BGCs) revealed by the continuously growing number of whole genome sequences included in public databases [6]. In fact, it is estimated that this phylum is responsible for the biosynthesis of two-thirds of all known antibiotics [7]. Many bioactive secondary metabolites have been isolated from actinobacteria employing the OSMAC strategy, which allows the expression of biosynthetic gene clusters that are commonly silent under standard laboratory fermentation conditions [4,8]. Since the discovery of actinomycin, streptothricin, and streptomycin in the 1940s, almost all major classes of antibiotics are produced by these bacteria. These include, among others, aminoglycosides as neomycin or kanamycin, ansamycins as rifamycin or geldanamycin, macrolides as erythromycin, glycopeptides as vancomycin or teicoplanin, β-lactams as cephamycinad, carbapenems as imipenem, and glycolipopeptides such as daptomycin [7].

Actinomycetes are widely distributed in natural environments, especially in soils, where they represent between 20 and 60% of the microbial population inhabiting plant rhizospheres or as part of the microbial communities associated with other living organisms [9,10]. Besides, several studies have revealed a rich biodiversity of actinobacteria in extreme environments such as deserts, permafrost soils, or deep-sea sediments. This biodiversity of adapted filamentous bacteria, which is still mostly uncharacterized, is considered an enormous potential source of new metabolites [11,12]. Therefore, these habitats constitute a suitable place for actinomycete bioprospection aiming for the identification of unknown species or genetically distinct strains and the discovery of new natural products with biological activities for clinical use as a treatment for microbial infections.

*El Saladar del Margen*, in Cúllar-Baza (Granada, Spain), is a wetland characterized by its climatic isolation exerted by nearby mountain ranges and by the consequent continentality and low rainfall. In addition, its geology is characterized by the salt accumulation in its soils because of the evaporation of phreatic mineralized water. These special climatic and geological conditions convert these wetlands into a high biodiversity hotspot with living organisms able to withstand the high salinity level of the substrate, thus resulting in an attractive area for bioprospecting.

Aiming to discover new antibiotic agents against multidrug-resistant microbes, a high-throughput-screening campaign with extracts from the actinobacterial strains isolated from *El Saladar del Margen* was carried out. For this purpose, the endemic plant *Limonium majus* was selected and collected from the halophilic substrate, where 108 actinobacteria strains were associated isolated and identified. The potential of these strains to produce new bioactive molecules was enhanced through an OSMAC approach by employing 10 different culture media in small-scale fermentations, and the subsequent antifungal and antibacterial activities of the extracts they produced were evaluated. Among the strains, *Kribbella* CA-293567 was highlighted because of the antibacterial and antifungal profiles that its extracts presented. Herein we describe its taxonomic identification and the bioassay-guided fractionation follow-up that drove us toward the chemical identification and activity characterization of the active components it produces.

## 2. Results and Discussion

### 2.1. Microbial Isolation and Identification of Actinobacterial Strains

A taxonomically diverse group of actinobacterial strains was isolated from the endemic plant *Limonium majus* and its associated rhizosphere, collected in *El Saladar del Margen*, in the Cúllar-Baza depression, Granada (Spain). A total of 108 actinobacterial strains were identified and classified based on their morphology and ribosomal 16S rDNA sequences. As a result, the following 9 different taxonomic orders, 15 families, and 25 genera of actinomycetales were identified: *Corynebacteriales*, *Micrococcales*, *Micromonosporales*, *Propionibacteriales*, *Pseudonocardiales*, *Streptomycetales*, *Streptosporangiales*, *Frankiales,* and *Kineosporiales* (Appendix A). The order *Streptomycetales* grouped the highest number of strains (*n* = 47), all from the genus *Streptomyces*, a widely studied group of actinobacteria, distributed in soils [13]. The second most representative order of the actinobacterial population was the *Micrococcales*, with five different families identified; *Brevibacteriaceae* (*n* = 3), *Microbacteriaceae* (*n* = 5), *Micrococcaceae* (*n* = 6), *Promicromonosporaceae* (*n* = 4) and *Dermabacteraceae* (*n* = 1), followed by the *Micromonosporales* order, with 2 families; *Micromonospora* (*n* = 17) and *Xiangella* (*n* = 1). However, eight actinomycete strains were classified as *Incertae sedis* and could not be assigned to any class.

Attending to their isolation origin in the *Limonium majus* plant, 72 of the 108 total strains were isolated from the rhizosphere, 26 from the roots, and only 10 from the aerial part. The soil surrounding the plant roots constitutes a rich source of nutrients called “rhizodeposits” due to root exudates, stimulating microbial growth. Therefore, this region is usually where the major microorganism diversity is located [9,14]. In contrast, due to the lack of nutrients, higher exposure to UV light, desiccation, and climatic changes, a lower number of microorganisms is generally isolated from the aerial parts, as observed in *Limonium majus* [15].

According to the 16S rDNA sequence of the producing strain, CA-293567 has a 99.41% sequence similarity to the strain *Kribbella koreensis* LM 161T (GenBank Accession No. Y09159), thus indicating the relatedness to this species. In addition, its phylogenetic position within the genus *Kribbella* was confirmed in the corresponding phylogenetic tree (Figure 1).

In contrast to other extensively studied actinobacteria, such as *Streptomyces*, a large number of minor genera and species that could have the potential to be the source of novel chemical entities of pharmaceutical interest have not yet been cultivated under laboratory conditions [3]. The *Kribbella* genus belongs to this group of less exploited actinobacteria, also referred to as “rare actinobacteria” [16]. This genus, which was originally identified as part of the genus *Nocardioides* in 1989 and reclassified in 1999 as a new genus [17,18], constitutes a potential producer of some new active antimicrobials.

Although the chemical potential of the genus has not been evaluated extensively to date, some species have shown a high potential for producing the following new bioactive agents: *Kribbella koreensis,* which produces neuropilin/growth factor complexes [19], *Kribbella antibiotica,* which shows antifungal activity [20], *Kribbella jejuensis* that inhibits the growth of *Streptomyces scabiei* [21], and *Kribbella* sp. UTMC 267 has been proven to possess anti-calcification properties [22]. In this context, the most remarkable bioactive compound described so far from this genus is sandramycin (**3**), a cyclic decadepsipeptide with strong antibacterial and antitumoral activity isolated from *Kribbella sandramycini* ATCC-39419 (Figure 2) [17,18,23].

### 2.2. Actinobacterial Production Screening: Small-Scale Fermentations

Each actinobacterial strain was small-scale cultured employing 96-deep well plates (0.8 mL in Duetz plates) and extracted, allowing a rapid evaluation of their bioactive profile potential.

Ten different liquid fermentation media were used, providing a range of nutritional conditions aimed to increase the chances of inducing the production of novel secondary metabolites through the OSMAC approach [24]. All extracts obtained from small fermentations were subsequently bio-assayed against a panel of microbial human pathogens, including Gram-positive (methicillin-resistant *Staphylococcus aureus*, MRSA MB-5393), Gram-negative (*A. baumannii* MB-5973, *Pseudomonas aeruginosa* MB-5919, *Escherichia coli* WT ATCC-25922 and *Klebsiella pneumoniae* 7006030) and a yeast strain (*C. albicans* ATCC-64124).

Out of the production screening, extracts of cultured strain CA-293567 in DNPM medium demonstrated very promising antibacterial activity against MRSA, *A. baumannii* MB-5973, *P. aeruginosa* MB-5919, and antifungal activity against *C. albicans* ATCC-64124, but no activity versus *E. coli* WT or *K. pneumoniae* 700,603 was observed.

In contrast, no antibiotic properties were detected for this strain in the other media, except against *C. albicans* ATCC-64124 and MRSA 5393 in APM-9 and DEF-15 media. These assay results gave further support to the extended “One strain/many compounds” strategy to induce the synthesis of secondary metabolites where the medium composition played a key role and confirmed the bioactive potential of the strain CA-293567 that was subjected to large-scale fermentations aimed at the isolation of its antibiotic components.

### 2.3. Isolation and Structural Elucidation of Compounds **1**–**5**

The strain *Kribbella* CA-293567 was fermented in two batches of 2 and 3 L of DNPM medium for 14 days at 28 °C, and culture broths and mycelia were extracted with acetone (1:1). The extracts were fractionated through a reverse-phase medium-pressure liquid chromatography and the resulting fractions were analyzed by LC-DAD-MS [24] and evaluated against a panel of pathogens as indicated in the methodology. Bioassay-guided fractionation was carried out using MRSA 5393, *P. aeruginosa* MB-5919, *A. baumannii* MB-5973, and *A. fumigatus* ATCC-46645, as general indicators of different antimicrobial activities.

After repeated sub-fractionation of the bioactive fractions and subsequent LC-DAD-HRMS analyses, the known compounds sandramycin (**3**), coproporphyrin III (**4**), and kribelloside C (**5**) were identified and purified in some of the bioactive fractions.

The LC-MS-DAD peaks with *m*/*z* values of 1221 and 1222 were related to sandramycin (**3**), which belongs to a class of *C*2-symmetric cyclic decadepsipeptides such as quinaldopeptin, luzopeptins, and quinoxapeptins and has proven to be a potent antibiotic agent [25]. In fact, this molecule, isolated originally from *Kribbella sandramycini* ATCC-39419, is also considered a potential antitumoral, antiviral, and antibacterial agent against Gram-positive bacteria [17,18,23,26]. It is worth noting that CA-293567 and *K. sandramycini* ATCC-39419 strains are distant *Kribbella* species in the phylogenetic tree (Figure 1). This may be explained by horizontal gene transference of natural product biosynthetic gene clusters, extensively observed with many other actinobacteria as an evolutionary mechanism [27]. Fractions containing sandramycin (**3**) displayed interesting activities against MRSA and *A. fumigatus*. Taking into account that no antifungal properties have previously been reported for **3**, it was completely isolated (7.6 mg) and confirmed by LC-HRESIMS ((+)-ESI-TOF) analysis, where [M + 2H]^2+^ and [M + NH_4_]^+^ ions at *m*/*z* 611.2822 and 1238.5844 (t_R_ = 5.92 min), respectively, were in agreement with a molecular formula of C_60_H_76_N_12_O_16_ (Appendix A). Purified sandramycin (**3**) was confirmed by NMR from one of the fermentation batches, resulting in a production yield of 2.53 mg/L.

Additionally, a peak with a *m*/*z* of 655 was dereplicated as coproporphyrin III (**4**) using our internal LC/MS database (Appendix A) [28]. This compound has a porphyrin core, whose antibiotic properties are based on its ability to catalyze peroxidase and oxidase reactions and generate reactive oxygen species [29]. Compound **4** has also recently been described as an interesting growth factor for uncultivable actinobacteria [30]. The molecule was not further characterized in our assay panel due to its widely described antimicrobial properties.

Finally, a bioactive compound with a molecular formula and an exact mass of C_31_H_58_O_14_ and 654.3825, respectively, known as kribelloside C (**5**), was also identified by LC-(+)-ESI-TOF analysis based on the presence of a [M + NH_4_]^+^ ion at *m/z* 672.4172 (t_R_ = 5.09 min) (Appendix A). This alkyl glyceryl ether, which does not display UV absorption, showed activity against *A. fumigatus*. It was originally isolated from *Kribbella* sp. MI481-42F6 and exhibited interesting activities as an antifungal agent versus *Saccharomyces cerevisiae* as an RNA 5′-triphosphatase inhibitor [31]. The compound was isolated, and its structure was identified by NMR. The sample obtained (4.7 mg from a 3 L fermentation; an average yield of 1.57 mg/L) was also characterized in our antimicrobial assay panel.

Interestingly, the LC-DAD-MS analyses also showed two non-dereplicated components with *m*/*z* signals of 768 and 782 in the bioactive fractions with interesting antimicrobial activities against *A. baumannii, P. aeruginosa,* and *A. fumigatus*. Consequently, these fractions were also pooled and analyzed by LC-HRESIMS (Appendix A), revealing the presence of two compounds with [M + H]^+^ ions at *m*/*z* 768.2312 (t_R_ = 1.08 min) and 782.2468 (t_R_ = 1.86 min), presenting molecular formulae of C_30_H_37_N_7_O_17_ and C_31_H_39_N_7_O_17_, respectively. No matches were obtained when searched against Fundación MEDINA’s high-resolution mass spectrometry databases and the commercial Chapman and Hall Dictionary of Natural Products.

The purification of these two molecules employing a semipreparative reversed phase yielded kribbellichelins A and B (**1** and **2**) (Figure 2; Appendix A) with an average yield production of 5.24 and 1.02 mg/L, respectively.

All identified active compounds produced by the strain are summarized in Table 1.

The molecular formula of C_30_H_37_N_7_O_17_ was assigned to kribbellichelin A (**1**) based on the presence of a protonated adduct in its (+)-ESI-TOF spectrum at *m/z* 768.2121. Analysis of its NMR data (Table 2) identified the presence in the molecule of signals corresponding to 8 carbonyls, 8 sp^2^ quaternary carbons, 2 sp^2^ methines, one oxygenated methylene, two methoxy groups, two nitrogenated methines, three nitrogenated methylenes, and four aliphatic methylenes, suggesting the presence of several amino acids in the structure. Indeed, the analysis of COSY and HMBC correlations (Figure 3a) identified two *β*-alanine, one serine, and one ornithine as structural elements of the molecule. The unusually deshielded chemical shift of C-19 in the latter (*d_c_* 48.9 in CD_3_OH) indicated a hydroxylation at N-18 similar to that occurring in fuscachelin B [32]. The MS/MS fragment at *m*/*z* 415.1460 (Figure 3b) also supports this proposal. Key correlations observed in its HMBC spectrum between H-15/C-13, H-19/C17, and H-22/C-25 determined the sequence β-Ala-N5-OH-Orn-Ser-β-Ala, which was confirmed by key fragments detected in the MS/MS analysis of the molecule (Figure 3b). The remaining signals in the NMR spectra accounted for two identical structural units, including two carbonyls, five sp^2^ carbons (one methine and four quaternary carbons), and one methoxy group. The HMBC correlations observed from H-4/H-33 of these moieties to C-2/C-36, C-5/C-32, and C-6/C-31 (Figure 3a) allowed them to be assigned a structure of methyl 6-carbonyl-4,5-dihydroxypicolinate. The ^13^C chemical shift values reported for a similar structural unit found in the *Streptomyces achromogenes* antibiotic rubradirin corroborated this proposal [33]. Both subunits were connected to carbons C-11 and C-27 of the β-Ala units as evidenced by HMBC correlations between H-11 and H-27 to C-9 and C-29, respectively, between NH-10 and C-9 and C-11, and between NH-28 and C-27 and C-29, observed in CD_3_OH (Figure 3a). Finally, the absolute configuration of the two chiral amino acid residues present in the molecule, Ser and Orn, was determined using Marfey’s analysis after hydrolysis of the molecule with HCl, which rendered the L configuration for serine or reductive hydrolysis with hydriodic acid [34], which removed the N5-hydroxy group from Orn and allowed it to establish its absolute configuration as L. For this aim, the retention times of the two hydrolyzed and derivatized aliquots of **1** were compared with the retention times of L and D standards of the amino acids Orn and Ser present in **1** derivatized with L-FDVA.

The (+)-ESI-TOF mass spectrum of kribbellichelin B (**2**) displayed a protonated adduct at *m*/*z* 782.2485 from which a molecular formula of C_31_H_39_N_7_O_17_ was assigned to the compound. This molecular formula contains an additional “CH_2_” unit with respect to **1.** The ^1^H and HSQC NMR spectra of **2** in CD_3_CN/D_2_O 1:1 (Table 2) were very similar to those of **1,** with the most remarkable difference being the presence of an extra methoxy signal (C-38) in the spectra of **2** at δ_H_/δ_c_ 3.57/53.5. Its location at carbonyl C-23 was secured through HMBC correlations from H_3_-38 and H-22 to C-23.

### 2.4. Biological Activity

Compounds **1**–**3** and **5** were tested against a panel of microbial human pathogens, including Gram-positive (MRSA MB-5393), Gram-negative bacteria (*A. baumannii* MB-5973 and *P. aeruginosa* MB-5919), and two fungi (*A. fumigatus* ATCC-46645 and *C. albicans* ATCC-64124), and evaluated for cytotoxicity versus the human hepatoma cell line HepG2 (ATCC HB-8065). Inhibition curves for positive results are shown in Figure 4.

Kribbellichelins A (**1**) and B (**2**) exhibited significant activity versus *C. albicans* with a measured IC_50_ of 11.7 µg/mL (15.2 µM) and 3.2 µg/mL (4.1 µM), respectively. Interestingly, both compounds showed an atypical inhibition profile against *A*. *fumigatus* where, as the concentration increased above an optimal fungicidal concentration, with a maximum of 40% inhibition at 7.2 µg/mL, less pathogen inhibition is observed at higher concentrations, with a decreasing inflection point close to 50 µg/mL. This paradoxical phenomenon, called the “Eagle effect”, was originally described for β-lactam antibiotics against Gram-positive bacteria [35,36], as well as the caspofungin marketed antifungal that also shows a similar behavior against *A. fumigatus* [36,37]. Weak inhibition activities were identified against Gram-negative *A. baumanii* and *P. aeruginosa*, with a maximum activity of growth inhibition plateau of 45% and 30%, respectively, at concentrations higher than 10 µg/mL. Finally, these two compounds showed no activity against Gram-positive MRSA 5393 nor significant cytotoxic activity against HepG2 at the concentration ranges where antimicrobial or antifungal activities were observed.

Sandramycin (**3**) showed a strong growth inhibition activity versus Gram-positive MRSA MB-5393, reaching an IC_50_ of 0.31 µg/mL (0.25 µM). Additionally, it exhibited remarkable antifungal activity against *A. fumigatus* ATCC-46645 and *C. albicans* ATCC-64124 with a measured IC_50_ of 5.72 µg/mL (4.69 µM) and 7.55 µg/mL (6.19 µM), respectively. In contrast, no antibacterial activity was identified for this compound against Gram-negative *A. baumannii* or *P. aeruginosa*. Previously, sandramycin (**3**) had only been proven to exert strong antibiotic activity versus Gram-positive bacteria like *Bacillus subtilis*, *S. aureus,* and *Streptococcus faecalis* [23]. Therefore, this is the first time its antifungal activity has been characterized. Sandramycin also displayed a remarkable cytotoxic activity against HepG2, reaching an ED_50_ of 1.578∙10^−3^ µg/mL (1.29 nM), five times higher than its Gram-positive MRSA MB-5393 potency value (0.25 µM). This cytotoxic activity is probably related to its two-fold axis of symmetry and two heteroaromatic chromophores that result in DNA binding properties characteristic of cyclic decadepsipeptides [26].

Finally, kribelloside C (**5**) displayed interesting antifungal activity against *A. fumigatus* with an IC_50_ of 13.65 µg/mL, in accordance with previously described antimicrobial activity against *S. cerevisiae*, but with no activity identified against the other pathogen strains of the panel, including the fungus *C. albicans* ATCC-64124 [31].

## 3. Materials and Methods

### 3.1. Microbial Isolation and Identification of Actinobacterial Strains

A taxonomically diverse subset of 108 actinobacterial strains (including the producer strain CA-293567) were isolated from the endemic plant *Limonium majus*, collected in salt soil in 2016 in *El Saladar del Margen*, in the Cúllar-Baza depression (Granada, Spain). Samples were air dried, heat-pre-treated, and suspended in sterile water. Suspensions were serially diluted, plated on selective isolation media, and incubated at 28 °C for at least 6 weeks. Strains were isolated from an NZ-amine-based agar medium containing nalidixic acid (20 µg/mL). Colonies were purified on Yeast Extract Malt Extract Glucose medium (ISP2) [yeast extract (4 g/L), malt extract (10 g/L), glucose (4 g/L), and agar (16 g/L)], adjusted to pH 7.2, and preserved as frozen agar plugs in 10% glycerol at −80 °C. These axenic strains are currently maintained in the Actinobacterial Collection of Fundación MEDINA (http://www.medinadiscovery.com, accessed on 1 August 2022). All reagents and medium components, unless specified, were of analytical grade and were obtained from Sigma-Aldrich (Merck Group, Darmstadt, Germany).

Genomic DNA was extracted from mycelium suspension grown on ATCC-2 medium 2 [soluble starch (20 g/L), dextrose (10 g/L), Sigma NZ amine EKC (5 g/L), Difco beef extract (3 g/L) (BD, Sparks, MD, USA), Bacto peptone (5 g/L), yeast extract (5 g/L), and CaCO_3_ (1 g/L), adjusted to pH 7.0 with NaOH before addition of CaCO_3_] [38]. DNA fragments containing the almost-complete 16S rRNA gene sequence (1350 nucleotides) were amplified with the FD1 (5′-AGAGTTTGATCCTGGCTCAG-3′) and RP2 (5′-ACGGCTACCTTGTTACGAC-3′) primers [39]. Reaction mixture preparation and PCR amplification were performed as described previously [40,41].

Amplification products were sequenced by Secugen (SECUGEN: Sequencing and molecular diagnostics, Madrid, Spain) (www.secugen.es, accessed on 1 August 2022). Partial sequences obtained of about 1300 nucleotides were assembled and edited with the Bionumerics^®^ software (6.6. version) (Applied Maths NV^TM^, Sint-Martens-Latem, Belgium), and subsequently compared with sequences deposited at EzBioCloud server (https://www.ezbiocloud.net, accessed on 1 August 2022) [42], and GenBank (https://www.ncbi.nlm.nih.gov/genbank, accessed on 1 August 2022). The 16S rRNA gene sequences of 96 out of 108 strain were deposited in GenBank for general public access under the following accession numbers: OP442265, OP442266, OP442267, OP442268, OP442269, OP442270, OP442271, OP442272, OP442273, OP442274, OP442275, OP442276, OP442277, OP442278, OP442279, OP442280, OP442281, OP442282, OP442283, OP442284, OP442285, OP442286, OP442287, OP442288, OP442289, OP442290, OP442291, OP442292, OP442293, OP442294, OP442295, OP442296, OP442297, OP442298, OP442299, OP442300, OP442301, OP442302, OP442303, OP442304, OP442305, OP442306, OP442307, OP442308, OP442309, OP442310, OP442311, OP442312, OP442313, OP442314, OP442315, OP442316, OP442317, OP442318, OP442319, OP442320, OP442321, OP442322, OP442323, OP442324, OP442325, OP442326, OP442327, OP442328, OP442329, OP442330, OP442331, OP442332, OP442333, OP442334, OP442335, OP442336, OP442337, OP442338, OP442339, OP442340, OP442341, OP442342, OP442343, OP442344, OP442345, OP442346, OP442347, OP442348, OP442349, OP442350, OP442351, OP442352, OP442353, OP442354, OP442355, OP442356, OP442357, OP442358, OP442359, and OP442360 (Appendix A).

In addition, phylogenetic and molecular evolutionary analyses of the strain CA-293567 were conducted using MEGA (6.06 version) [43]. Multiple alignments were carried out using CLUSTALX [44], integrated into the software. The phylogenetic analysis was based on the Neighbor-Joining method using matrix pairwise comparisons of sequences corrected with Jukes and Cantor algorithm [45,46].

### 3.2. General Experimental Procedures

The 1-D and 2-D NMR spectra were recorded at 297K on a Bruker Avance III spectrometer (500 and 125 MHz for ^1^H and ^13^C NMR, respectively) equipped with a 1.7 mm TCI MicroCryoProbe^TM^ (Bruker Biospin, Fällanden, Switzerland). The ^1^H and ^13^C chemical shifts were reported in ppm using the signals of the residual solvents as internal reference (δ_H_ 3.31 and δ_C_ 49.1 ppm for CD_3_OH; δ_H_ 1.93 and δ_C_ 1.3 ppm for CD_3_CN). LC-UV-LRMS analyses were performed on an Agilent 1100 single quadrupole LC-MS system (Agilent Technologies, Santa Clara, CA, USA) as previously described [28]. LC-HRESIMS ((+)-ESI-TOF) mass spectra were acquired using a Bruker maXis QTOF mass spectrometer (Bruker Daltonik GmbH, Bremen, Germany) coupled to an Agilent Rapid Resolution 1200 HPLC, and dereplication was performed as described previously [28,47].

Medium-pressure liquid chromatography (MPLC) was performed on a semiautomatic flash chromatography system CombiFlash from Teledyne ISCO Rf 200 (Teledyne ISCO, Lincoln, NE, USA) with an SP207ss resin column. Preparative and semipreparative HPLC separation was performed on a Gilson GX-281 322H2 (Gilson Technologies, Middleton, WI, USA) with preparative and semipreparative reversed-phase columns (Zorbax SB-C_8_, 21.2 × 250 mm, 7 μm, and Zorbax RX-C_8_, 9.4 × 250 mm, 5 μm, respectively). Acetone used for extraction was analytical grade. Solvents employed for isolation were all HPLC grade. Chemical reagents and standards were purchased from Sigma-Aldrich.

### 3.3. Small Fermentations and Extraction

To conduct the small fermentation screenings, the strains’ inocula were prepared as follows: A first seed of the actinomycetes was carried out by inoculating 10 mL of seed medium ATCC-2 in a 10 mL tube with 0.5 mL of a frozen inoculum stock of the strains and incubating the tubes at 28 °C for about 48 h shaking at 220 rpm and 70% of humidity.

Small-scale liquid fermentations in 0.8 mL cultures of the actinobacterial strains were performed employing 96-deep well plates (Duetz system, www.enzyscreen.com, accessed on 1 August 2022) [48,49,50,51]. Ten different culture media (APM-9, DEF-15, DNPM, FPY-12, -FR23. GPA. KHC, M016, RAM2-P V2, and SAM-6, all at 28 °C and 75% of humidity) were used in 7 days cultures for *Streptomyces* and in 14 days cultures for the remaining strains [24]. After incubation, 0.8 mL of acetone was added to each culture. The mixture was shaken for 3 h at 200 rpm and then centrifuged (Rotanta 460 RS Hettich Zentrifugen, Bremen, Germany) to discard the mycelial debris. The acetone extracts were then evaporated in a GeneVac HT-24 (SP Industries, Warminster, Pennsylvania, USA) centrifugal evaporator. Finally, the precipitate was dissolved in 200 µL of 20% DMSO and added to an AB-0765 ABgene™ plate for antimicrobial evaluation.

Those strains yielding bioactive extracts were confirmed at a higher volume in duplicates in EPA vials systems of 40 mL. These medium-scale fermentations of 10 mL were grown for 7 and 14 days at 28 °C, 75% of humidity, and 220 rpm. Next, 10 mL of acetone was added to each culture and mixed for 1 h and then evaporated under nitrogen steam down to 9 mL removing acetone in presence of 20% final DMSO.

### 3.4. Large-Scale Fermentation, Extraction, and Isolation

The producing strain CA-293567 inoculum was prepared as indicated for small-scale fermentations. A second seed culture was prepared by inoculating 50 mL of seed medium ATCC-2 in two 250 mL baffled flasks with 2.5 mL of the first seed, which were incubated at 28 °C for 5 days shaking at 220 rpm and 70% of humidity. A 5% aliquot of the second culture was transferred to each of the 500 mL Erlenmeyer flasks containing 150 mL of the production medium DNPM [Type I corn dextrin (Sigma) (40 g/L), N-Z Soy BL (Sigma) (7.5 g/L), Bacto yeast extract (Difco) (5 g/L), MOPS (Fisher) (21 g/L), pH = 7.0]. Erlenmeyer flasks were incubated at 28 °C for 7 days shaking at 220 rpm and 70% of humidity.

Whole broths of 2 and 3 L were extracted with an equal volume of acetone under continuous shaking at 220 rpm for 3 h. The resulting combined extract was centrifugated at 7500 rpm for 10 minutes and vacuum filtered over paper in Büchner to discard the mycelial debris. Finally, the organic solvent was evaporated under an N_2_ steam to a final volume of 1.3 L (100% water).

The aqueous residues were loaded onto reverse-phase SP207ss resin columns (RediSep^®^, Lincoln, NE, USA, 65g, 100 × 35 mm) that were eluted with a stepwise Acetone:H_2_O gradient (10 mL/min flow; 10–100% acetone in 42 min, increasing 20% each 6 min; 20 mL/fraction), yielding 24 fractions. After antibiotic evaluation, fractions containing the active compounds were pooled into three groups according to their bioactivity results and LC-UV-MS profiles and evaporated to dryness in a centrifugal evaporator GeneVac HT-24 to yield fraction pools FS013 and FS018. Both pools were further chromatographed by preparative reversed-phase HPLC column (Zorbax SB-C8, Agilent Technologies, Santa Clara, CA, USA, 21.2 × 250 mm, 7 μm; 20 mL/min; UV detection at 210 and 280 nm).

Pool FS013 was chromatographed applying a linear CH_3_CN:H_2_O gradient (5–35%: 1–34 min, 35–100%: 35–36 min, 100%: 36–43 min), both solvents containing 0.1% TFA.

Pool FS018 was also chromatographed employing a linear CH_3_CN:H_2_O gradient (5–100%: 1–36 min, 100%: 36–43 min), both solvents containing 0.1% TFA.

Fractions from FS013 and FS018 chromatographies containing kribbellichelins A (**1**) and B (**2**), as confirmed by LC-MS analyses, were pooled and further re-purified by semi-preparative reversed-phase HPLC (Zorbax RX-C8, 9.4 × 250 mm, 5 μm) applying a linear H_2_O:CH_3_CN gradient (5–35%:1–34 min, 35–100%: 35–36 min, 100%: 36–43 min). The separation yielded the purified compounds **1** (11.6 and 14.6 mg from 2 to 3 L fermentations, respectively, t_R_ 26 min) and **2** (0.4 and 4.7 mg, t_R_ 28.5 min) as clear-white amorphous wax solids.

Sandramycin (**3**) (7.6 mg, t_R_ 27.5 min), coproporphyrin III (**4**) (1.3 mg, t_R_ 15.5 min), and kribelloside C (**5**) (4.7 mg, t_R_ 24 min) were identified and isolated only from FS018 pool purification as a result of LC-MS dereplication and comparison against Fundación MEDINAs high-resolution mass spectrometry database.

### 3.5. Marfey’s Analysis of Compound **1**

A sample (200 µg) of compound **1** was divided into two portions that were dissolved in 0.3 mL of 6 N HCl or 0.3 mL of 6N HI and heated at 110 °C for 16 h. The resulting mixtures were dried overnight under an N_2_ stream. The solid residues were resuspended in 50 µL of a 1 M NaHCO_3_ solution that turned violet. Once the solutions stopped fizzing, 150 µL of L-FDVA (Marfey’s reagent, N-(2,4-dinitro-5-fluorophenyl)-L-valinamide) was added to each of the solutions. The reaction mixture was heated at 40 °C for 40 min until the solutions turned red. After that time, the reactions were quenched by dropwise addition of 1 N HCl until the mixtures turned yellow again. For the HPLC analysis, 10 µL of the derivatives solution was added to 40 µL of acetonitrile and analyzed by LC/MS on an Agilent 1260 Infinity II single quadrupole LC/MS instrument. Separations were carried out on a Waters X-Bridge C18 column (4.6 × 150 mm, 5 um) maintained at 40 °C. A mixture of two solvents, A (10% acetonitrile, 90% water) and B (90% acetonitrile, 10% water), both containing 1.3 mM trifluoroacetic acid and 1.3 mM ammonium formate, was used as the mobile phase under a linear gradient elution mode (25−65% B in 28 min, 65–100% B in 0.1 min, then isocratic 100% B for 4 min) at a flow rate of 1.0 mL/min. The amino acid standards were derivatized and analyzed following the same methodology described for the hydrolizates of compound **1**. HPLC traces of these analyses are shown in Appendix A. Retention times (min) for the observed peaks in the HPLC trace of the L-FDVA-derivatized amino acid standards were as follows: L-Ser: 5.58 min, D-Ser: 6.07 min, D-Orn: 13.94 min, and L-Orn: 14.98 min. Retention times (min) for the observed peaks in the HPLC trace of the L-FDVA-derivatized hydrolysis product of compound **1** were as follows: L-Ser: 5.37 min, and L-Orn: 14.97 min.

### 3.6. Antimicrobial Assays

Purification fractions and purified compounds **1**–**3** and **5** were tested against a wide panel of antimicrobial assays for the growth inhibition of human pathogens including MRSA MB-5393, *A. baumannii* MB-5973, *P. aeruginosa* MB-5919, *A. fumigatus* ATCC-46645, and *C. albicans* ATCC-64124 as 10-point curves with 1:2 dilutions starting at 128 μg/mL in duplicate, following previously described methodologies [52,53,54,55,56]. Most of the pathogens were acquired from the American Type Culture Collection (ATCC)(Manassas, VA, USA), except for those already present in our MEDINA bacteria collection (MB). Pathogen panel investigated was selected according to typical reference strains and procedures indicated by the NCSLI (National Conference of Standards Laboratories International) (Boulder, CO, USA).

Gram-negative antibacterial assay was performed following this procedure; thawed stock inoculum from vials was streaked onto MHII agar plates and incubated overnight at 37 °C. Isolated colonies were selected and inoculated in 25 mL of MHII liquid growth medium in 250 mL Erlenmeyer flasks overnight at 37 °C. MHII liquid growth medium was used in a 96-well plate assay, adjusting at 5 × 10^5^ colony forming units (cfu)/mL (OD600 nm) [56].

For Gram-positive MRSA, a thawed stock inoculum suspension from a cryovial of this microorganism was streaked onto brain heart infusion (BHI) agar plate and incubated at 37 °C overnight to obtain isolated colonies. Single colonies were inoculated into 10 mL of BHI broth medium in 250 mL Erlenmeyer flask and incubated overnight at 37 °C with shaking at 220 rpm and then diluted in order to obtain an assay inoculum of 1.1 × 10^6^ CFU/mL [52].

The antifungal assay was performed following this methodology; briefly, a thawed stock inoculum suspension from a cryovial of *A. fumigatus* and *C. albicans* was streaked onto PDA and Sabouraud Dextrose Agar medium, respectively, and suspended in RPMI-1640 modified medium and incubated at 37 °C for 24 h. Spores concentration of *A. fumigatus* was determined by counting in Neubauer chamber in order to dilute the culture and obtain an assay inoculum of 2.5 × 10^4^ spores/mL. For *C. albicans*, the OD660 was adjusted to 0.25 using RPMI-1640 modified as diluent and blank. Then, a dilution from the adjusted inoculum was prepared at 1/100 to inoculate 96-well assay plates [53,54].

For the assay, 90 μL/well of the diluted inoculum was mixed with 1.6 μL/well of compound dissolved in DMSO and 8.4 μL/well of MHII or BHI medium for Gram-negative and Gram-positive strains, respectively. Vancomycin was included as an internal plate control for MRSA. A rifampicin, aztreonam, and amphotericin B dose-response curve was used as a positive control against *P. aeruginosa*, *A. baumannii* and *A. fumigatus,* and *C. albicans*, respectively. Each compound was tested as a 10-point dose-response curve by 1:2 serial dilutions in DMSO starting at 128 µg/mL in duplicate. Total growth was measured using the EnVision^®^ (Redwood City, CA, USA) Multilabel plate reader (Perkin Elmer) with two readings, one at initial time and one after which the microorganism was incubated at 37 °C for 18–20 hours. For the *A. fumigatus* assay, total growth was measured with a single reading after the incubation [55].

In order to process and analyze the data and calculate the RZ’ factor (which predicts the robustness of an assay), the Genedata Screener software (Genedata, Inc., Basel, Switzerland) was employed. In all experiments performed in this work, the RZ’ factor obtained was between 0.85 and 0.92. Prim 9.4.1 software, from GraphPad was used for titrations representation [56].

### 3.7. Cytotoxicity Bioassays

Pure compounds were assayed against the human hepatoma cell line HepG2 (ATCC HB-8065) in an MTT test as 20-point curves with 1:2 dilutions starting at 40 μg/mL in triplicate. Cells were seeded at 10.000 cells/well in a 96-well plate (Corning 96-well TC-treated microplates) for 24 h and after addition of compounds **1–3** and **5** plates were incubated for 72 h. MMS (methylmethanesulfonate, Sigma-Aldrich, 4 mM) was used as the positive control, and DMSO 0.5% as the negative control. After addition of MMT dye (thiazolyl blue tetrazoliumbromide, ACROS Organics), cells were incubated for 2–3 h and supernatant was removed. Resulting formazan crystals were finally dissolved by means of 100 µL DMSO (100%) and absorbance was measured at 570 nm. The obtained data was analyzed using Genedata Screener software (Genedata, Inc., Basel, Switzerland) and Prim 9.4.1 software, from GraphPad was used for titrations representation [57].

## 4. Conclusions

Two new natural products, kribbellichelins A (**1**) and B (**2**), were isolated along with sandramycin (**3**), coproporphyrin III (**4**), and kribelloside C (**5**) from cultures of the halophilic strain *Kribbella* CA-293567, genetically closely related to *Kribbella koreensis*. Both compounds displayed modest antibiotic activity values versus clinically relevant yeast *C. albicans* (IC_50_ of 11.72 µg/mL (15.28 µM) and 7.55 µg/mL (9.66 µM), respectively). On the contrary, both compounds showed minor inhibitory properties against Gram-negative bacteria *A. baumannii* and *P. aeruginosa*, and no activity versus MRSA. In any of the cases, compounds presented MICs for total growth inhibition of the antimicrobial agents evaluated. Their structural resemblances to known siderophores may indicate a possible siderophore-nature-related mechanism of action to further evaluate. In contrast, sandramycin (**3**) showed excellent antibiotic activities against MRSA, *A. fumigatus,* and *C. albicans* (IC_50_ of 0.31 µg/mL (0.25 µM), 5.72 µg/mL (4.69 µM) and 7.55 µg/mL (6.19 µM), respectively), constituting this as the first report on its antifungal properties. Moreover, kribelloside C confirmed interesting antifungal properties versus *A. fumigatus* with an IC_50_ of 13.65 µg/mL, not observed against the fungus *C. albicans*.

Finally, compounds **1, 2,** and **5** showed no relevant cytotoxicity against HepG2 in contrast to sandramycin (**3**), which displayed a significant ED_50_ value (ED_50_ = 1.578∙10^−3^ µg/mL; 1.29 nM). Therefore, the two new kirbellichellins (**1** and **2**), sandramycin (**3**), and kribelloside C (**5**) can explain the antimicrobial profile observed for the extracts of *Kribbella* CA-293567 fermentations and can be added to the plethora of different natural product structures available as starting points for developing new antibiotics against drug-resistant microbes.

## Figures and Tables

**Figure 1 molecules-27-06355-f001:**
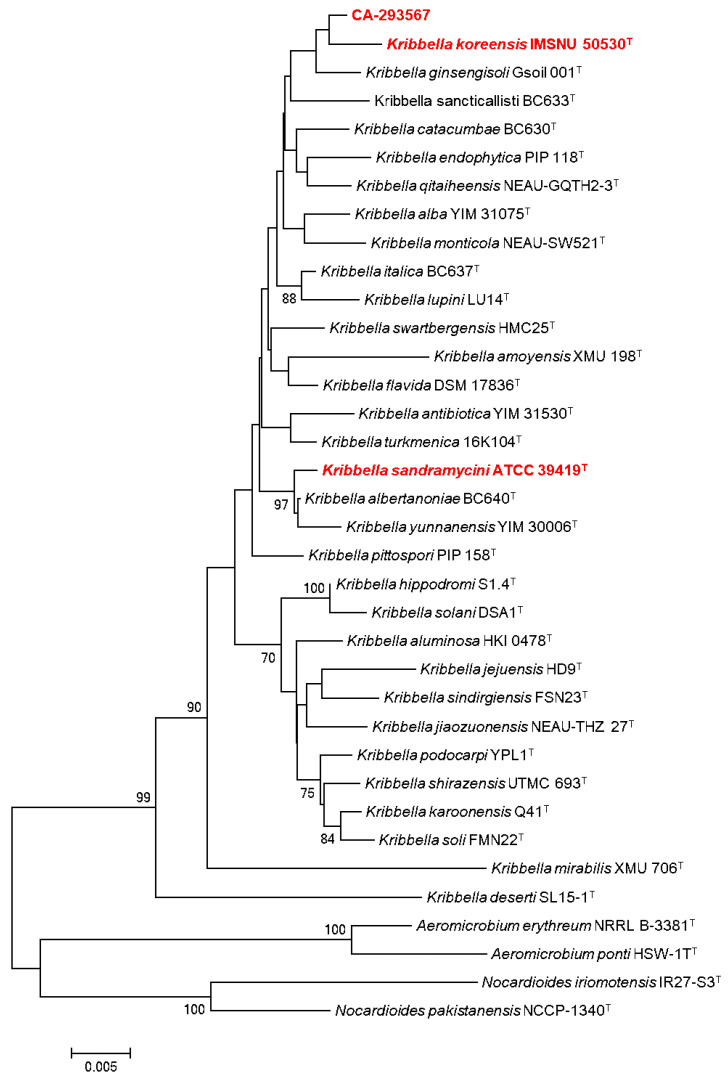
Phylogenetic tree based on the 16S rRNA partial gene sequences (≈1351 nt) illustrating the position of the strain CA-293567 among recognized species of the genus *Kribbella* and other related taxa. Determined by neighbor-joining method (Jukes and Cantor evolutive model) using MEGA (6.06 version) software. The numbers above the branches indicate the bootstrap values.

**Figure 2 molecules-27-06355-f002:**
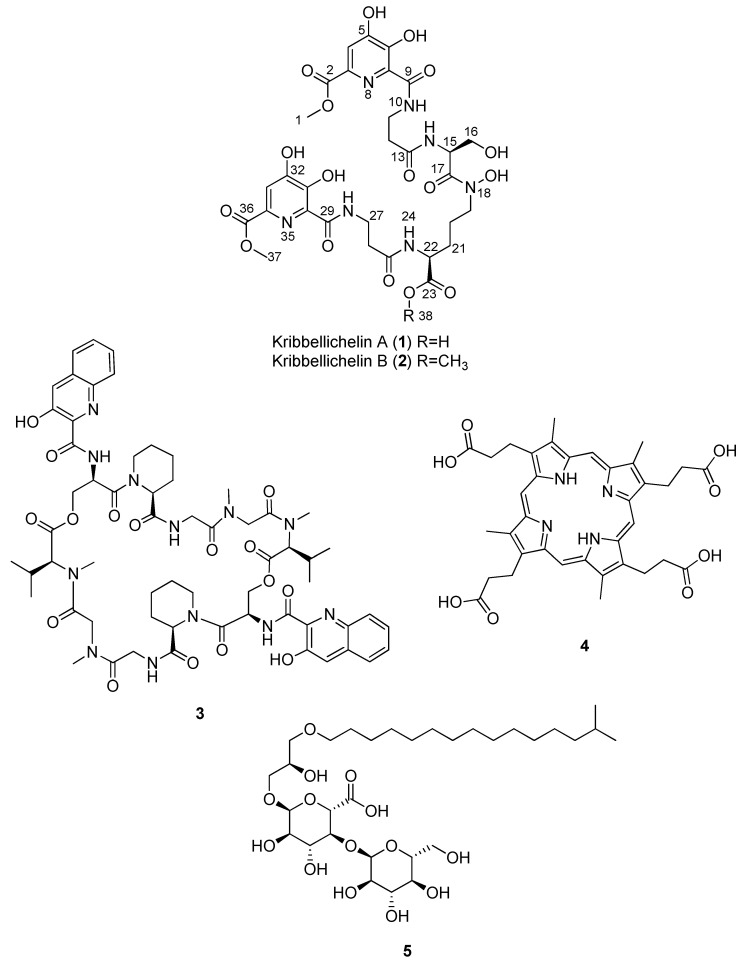
Structures of kribbellichelins A (**1**) and B (**2**), sandramycin (**3**), coproporphyrin III (**4**), and kribelloside C (**5**).

**Figure 3 molecules-27-06355-f003:**
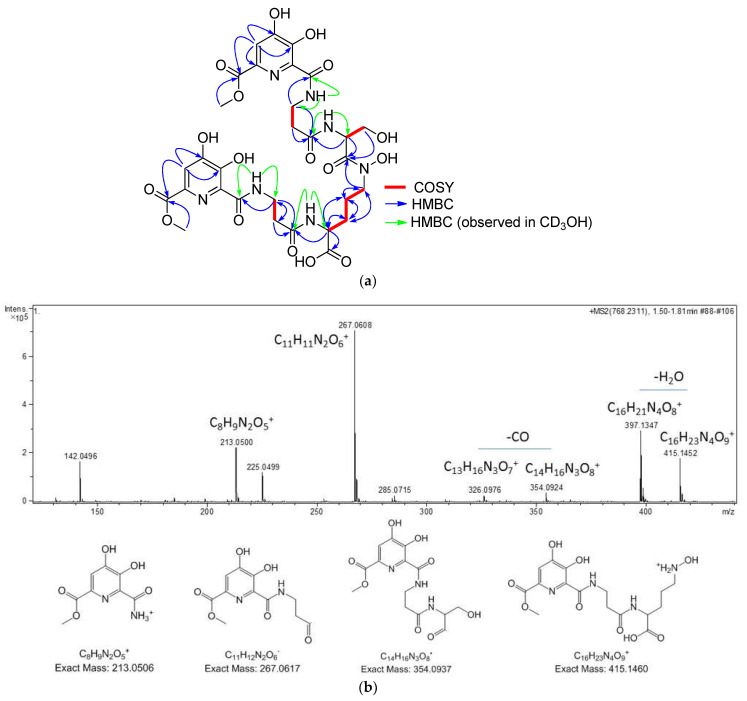
(**a**) Key COSY and HMBC correlations observed in the structure of kribbellichelin A (**1**). (**b**) MS/MS fragmentation of kribbellichelin A (**1**).

**Figure 4 molecules-27-06355-f004:**
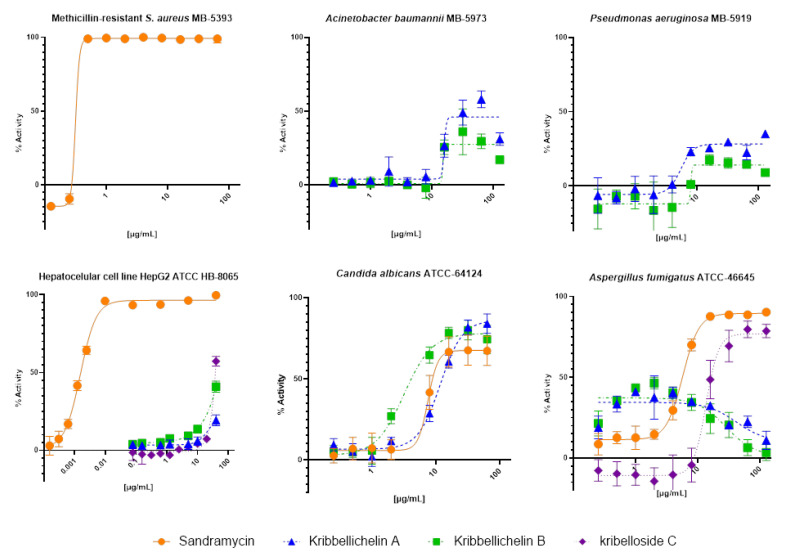
Antimicrobial/cytotoxic activity of compounds **1**–**3** and **5** against MRSA MB-5393, *A. baumannii* MB-5973, *P. aeruginosa* MB-5919, HepG2 (ATCC HB-8065), *A. fumigatus* ATCC-46645, and *C. albicans* ATCC-64124.

**Table 1 molecules-27-06355-t001:** Summary of identified active compounds **1**–**5**.

Compound	MS Signals (*m*/*z*)	Exact Mass (Da)	Molecular Formula
Kribbellichelin A (**1**)	768	767.2246	C_30_H_37_N_7_O_17_
Kribbellichelin B (**2**)	782	781.2402	C_31_H_39_N_7_O_17_
Sandramycin (**3**)	1221, 1222	1220.5502	C_60_H_76_N_12_O_16_
Coproporphyrin III (**4**)	655	-	C_36_H_38_N_4_O_9_
Kribelloside C (**5**)	672	654.3825	C_31_H_58_O_14_

**Table 2 molecules-27-06355-t002:** NMR data (500 MHz) of compounds **1** and **2**. ^13^C chemical shifts were obtained from HSQC and HMBC experiments. Coupling constants (*J*) indicated in Hz.

	1				2	
	CD_3_OH		CD_3_CN/D_2_O 1:1		CD_3_CN/D_2_O 1:1	
	^1^H NMR	^13^C NMR	^1^H NMR	^13^C NMR	^1^H NMR	^13^C NMR
Position	*δ* in ppm, mult, *J* in Hz	*δ* in ppm	*δ* in ppm (mult, *J* in Hz)	*δ* in ppm	*δ* in ppm (mult, *J* in Hz)	*δ* in ppm ^j^
1	3.94, s	53.5	3.83, s	54.0	3.85, s	54.0
2		166.9		166.40 ^c^		166.6
3		139.2		138.46 ^d^		n.d.
4	7.54, s	116.1	7.45, s	115.80 ^e^	7.52, br s	115.8
5		155.8		155.6		n.d.
6		151.9		150.9		n.d.
7		132.3		131.33 ^f^		n.d.
8						
9		170.6		169.2		169.2
10	9.34, m					
11	3.69, m, 2H	37.0 ^a^	3.55, m, 2H	36.52 ^g^	3.57, m, 2H	36.5
12	2.64, m, 2H	36.4	2.56, m, 2H	35.81 ^h^	2.55, m, 2H	35.8
13		173.73 ^b^		173.94 ^i^		173.8
14	8.05, d, 7.5					
15	5.12, m	53.8	4.96, m	53.2	4.95, m	53.2
16	3.81, dd, 11.0, 4.4; 3.73, dd, 11.1, 6.2	62.6	3.70, dd, 11.3, 4.5; 3.63, dd, 11.3, 6.3	61.7	3.68, m; 3.62, m	61.7
17		171.5		171.1		171.2
18						
19	3.57, m, 2H	48.9	3.51, m; 3.39, m	48.6	3.48, m; 3.37, m	48.7
20	1.67, m, 2H	24.3	1.53, m, 2H	23.5	1.49, m, 2H	23.5
21	1.84, m; 1.63, m	29.7	1.69, m; 1.56, m	28.7	1.65, m; 1.51, m	28.8
22	4.40, m	53.7	4.22, m	53.3	4.24, m	53.4
23		175.4		175.7		174.5
24	8.30, d, 7.7					
25		173.65 ^b^		173.85 ^i^		174.0
26	2.64, m, 2H	36.4	2.56, m, 2H	35.65 ^h^	2.55, m, 2H	35.8
27	3.69, m, 2H	36.9 ^a^	3.55, m, 2H	36.40 ^g^	3.57, m, 2H	36.5
28	9.34, m					
29		170.6		169.2		169.2
30		132.3		131.29 ^f^		n.d.
31		151.9		150.9		n.d.
32		155.8		155.6		n.d.
33	7.54, s	116.1	7.45, s	115.76 ^e^	7.52, brs	115.8
34		139.2		138.41 ^d^		n.d.
35						
36		166.9		166.36 ^c^		166.6
37	3.94, s	53.5	3.83, s	54.0	3.83, s	54.0
38	-		-		3.57, s	53.5

^a, b, c, d, e, f, g, h, i, j^ Interchangeable assignments.

## Data Availability

The data presented in this study are available in Appendix A.

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
