# Peer review of "Kribbellichelins A and B, Two New Antibiotics from Kribbella sp. CA-293567 with Activity against Several Human Pathogens"

_molecules, 2022, doi:10.3390/molecules27196355_

Round 1

Reviewer 1 Report

The paper's authors isolated five metabolites from actinobacteria through an interesting approach by changing culture mediate. They screened against one Gram-positive, two Gram-negative, and two fungi microbial human pathogen. Most significantly, two new natural compounds were discovered. The structure determination for Kribbellichelin A and B (please check the spelling in Figure 2) is reasonable, although X-ray crystallography could be used for further confirmation. The bioactivity study shows some activity towards two Gram-negative bacteria strains and strong antifungal activity against C. albicans ATCC-64124, which would be a great starting point for following medicinal chemical modification. But the abnormal behavior against A. FumigatusATCC-46645 could be a rising concern for considering those two compounds as potential antifungal candidates. A systematic mechanism study would be expected in future research. The authors also discovered the antifungal activity of Sandramycin, a known metabolite, for the first time. The authors claimed the antitumoral activity for Sandramycin based on the toxicity towards one cell line without knowing the drug-action mechanism or its effects on prime cells is not appropriate. Overall, this paper has a meaningful impact on future drug discovery. I suggest it be accepted with minor revision.

Author Response

Authors are grateful for the comments and considerations indicated by the Reviewer.

As indicated, spelling of Figure 2 has been corrected.

We agree in the comment that X-ray crystallography might be used for further confirmation of the structure, but the amount of material isolated did not allow to obtain crystals suitable for that purpose. Nevertheless, we are confident that the approach used to determine the structures of the new compounds is solid enough to unequivocally assign the structure and absolute configuration of the molecules.

Authors also agree that presented results for Sandramycin only indicates its toxicity towards an immortal Hepatocellular cell line (HepG2) and that it cannot be concluded an antitumoral activity. Authors have changed to cytotoxicity any discussion on the presented results and have left the antitumoral possibility to the bibliography where that potential had been indicated.

Again, thank you for the constructive points indicated.

Author Response

Authors are thankful for the improvements to the work indicated by the referee. And have address the whole text by introducing briefly the abstract and trying the introduction to be more focused and present recent references. Pathogens sources and its selection has been added to the text.

Accession numbers for the sequences in the study have been upload to GenBank and are referenced.

Pathogens’ selective media, antimicrobial standards, as well as, the references for all methods have been checked to be indicated in the text.

Authors consider results are correctly depicted in the different tables and figures of the text and the supplementary information, please indicate us if the referee feels the lack of a specific table or figure for the authors to add it.

A native English speaker has reviewed the text to check the correct structure of the sentences and the use of abbreviations.

Statistical analyses of the activity profiling of the different compounds have been determined using the Genedata Screener software and titration graphics depicted by using GraphPad commercial software, the statistical process of the data is depicted in more detail in the references for each assay (references 51 to 56).

Again, thank you very much for the constructive points indicated.

Reviewer 3 Report

This manuscript describes the detailed screening of the strains, following by the small-scale fermentations, extraction, isolation, structure elucidation, and antimicrobial and cytotoxicity bioassays of the secondary metabolites from the cultures of Kribbella sp. CA-293567 strain. It is interesting to find two new compounds consisting of Ala, N5-OH-Orn, Ser, Ala and methyl 6-carbonyl-4,5-dihydroxypicolinate. And Marfey’s method was used for the determination of the absolute configuration of their amino acid residues. Moreover, they were evaluated antimicrobial activity against a panel of relevant antibiotic-resistant pathogenic strains and cytotoxic activity versus the human hepatoma cell line HepG2. It is worthy to point out that the significant antimicrobial activity with low cytotoxicity make these structures available as starting points for developing new antibiotics against drug-resistant microbes. However, revisions are required if the article is considered for publication.

Q1: The compound 4 was preliminary identified using the internal LC/MS database, but there was no LC-MS chromatogram in the Supplementary Materials. Please add it.

Q2: Please add the 13C and 1H-1H COSY spectra of compound 2 in the Supplementary Materials. And please modify the HSQC and HMBC spectra. It is better to add figure legends in the supplementary document.

Q3: The mass data ‘m/z 768.2323’ of compound 1 on P7L237 was not consistent with that shown in Figure S6. Please revise it.

Q4: What are ‘two nitrogenated methines’ and ‘three nitrogenated methylenes’ in the structure of compound 1? There are seven aliphatic methylenes, not four aliphatic methylenes on P7L241. Moreover, there are two aliphatic methines but not mentioned in the manuscript.

Q5: The 1H-1H COSY correlation of H-15 (δH 5.18)/H2-16 (δH 3.81, 3.73) could be observed in Figure S9, but was not shown in Figure 3a.

Q6: ‘The unusually deshielded chemical shift of C-19 in the latter (dc 48.9 in CD3OH) indicated a hydroxylation at N-18.’ Please add a reference compound here. And was there any direct evidence for the hydroxylation at N-18, for example, IR?

Q7: For the subunit methyl 6-carbonyl-4,5-dihydroxypicolinate, there was only one methoxy group, not two on P7L249. Although the gross carbon framework could be determined by the HMBC correlations, there is lack of evidence for the ortho disubstitution of two hydroxyl groups. ‘13C chemical shift values reported for a similar structural unit found in the Streptomyces achromogenes antibiotic rubradirin corroborated this proposal [32].’ Please give a detailed interpretation for this.

Q8: What are the positive controls for the antimicrobial and cytotoxicity bioassays? Please add the IC50 valuses reported in the µM for ‘11.7 µg/mL and 3.2µ g/mL’ on P10L289 like ‘0.31 µg/mL (0.25 µM)’ on P11L309.

1. P1L15:  ‘this strain’   ‘the strain Kribbella sp. CA-293567’

2. P1L20:  ‘a bioassay guided fractionation’   ‘a bioassay-guided fractionation’

3. P2L68:  ‘…cephamycinad carbapenems as imipenem…’   …cephamycinad, carbapenems as imipenem…

4. P3L119:   ‘…called “rhizodeposits” thanks to root exudates, stimulating microbial growth’ This sentence is obscure.

5. P3L141:   ‘…has proven to possess anti-calcification properties’   …has been proven to possess anti-calcification property

6. Figure 1 caption:  ‘value’   values

7. P6L189:  ‘…and constitutes one of the most important previously isolated compounds from Kribbella genus.’ This sentence is obscure.

8. P6L199:  ‘[M+2H]+2  ‘[M+2H]2+

9. Figure S2 caption:  5% CH3CN  5% CH3CN

10. Figure S4 caption:  compounds 5  compound 5

11. Figure S5 caption:  5% CH3CN  5% CH3CN

12. P6L216:  ‘The compound was isolated and, after confirming its structure by NMR, the sample obtained…’   ‘The compound was isolated and its structure was identified by NMR. The sample obtained…’

13. Figure 2 caption:  ‘Structure of…’   ‘Structures of…’

14. P6L228:  ‘Figures S6 and S7’   ‘Figures S6 and S17’

15. P7L238:  ‘its NMR spectra’   ‘its NMR data’

16. P7L246:  ‘H22/C-25’   ‘H-22/C-25’

17. Figure 3b:  The subscript format of the numbers in the molecular formula C11H11N2O6.

18. P8L269:  dH/dc 3.57/53.5’   δH/δC 3.57/53.5’

19. P8L270:  ‘H-38 and H-22 to C-23’   ‘H3-38 and H-22 to C-23’

20. Please move the Conclusions after the Materials and Methods. And please add the physical and chemical properties of the compounds after the isolation procedure on P14, especially for the two new compounds.

Author Response

Authors are thankful for the improvements to the work indicated by the referee.

Q1: Compound 4 LC-UV-MS dereplication details have been added to the Supplementary Information.

Q2: Please note that due to the structural similarity between compounds 1 and 2, the COSY spectrum was not recorded due to the lack of useful information that could be extracted from it. The 13C NMR spectrum was not recorded due to scarcity of sample and  13C NMR shifts were deduced from HSQC and HMBC spectra. This is now indicated in table 2 as a footnote.

Q3: This is correct. We have replaced 768.2323 by 768.2121

Q4: Nitrogenated methines and nitrogenated methylenes are methines and methylenes attached to a nitrogen atom. We believe that the number of nitrogenated methines/methylenes and aliphatic methylenes (attached only to carbon atoms) indicated in the article is correct.

Q5: Thanks for pointing this out. We have modified the figure accordingly.

Q6: We have replaced the sentence by the following and added a reference: “The unusually deshielded chemical shift of C-19 in the latter (dc 48.9 in CD3OH) indicated a hydroxylation at N-18 similar to that occurring in fuscachelin B. The MS/MS fragment at m/z 415.1460 (Fig.3b) also supports this proposal.“ Please note that due to the existence of other hydroxy groups in the molecule IR spectroscopy is not considered an appropriate technique for the confirmation of the presence of a OH  group at N-18.

Q7: We have modified the text according to your comment. Please note that in a compound with a meta disposition of the two hydroxy groups, the sp2 methines H-5/H-32 would be 4 bonds apart from C2/C-35, making questionable the existence of an intense HMBC correlation between these pairs observed in figure S11. Please note that we have replaced H-3 by H-4 in P7L250 due to an error in the previous version of the manuscript. Finally, reference 32 describes the structure of rubradirin, a compound with a similar 6-carbonyl-4,5-dihydroxypicolinate subunit whose 13C chemical shifts, although slightly different from those observed in our compound due to a different substitution pattern, corroborate the structural proposal.

Q8: Controls for the antimicrobial and cytotoxicity bioassays have been reflected in the M&M section

The indicated list of other typos, corrections, changes in the order and suggestions have been addressed in the text

Again, thank you for the constructive points indicated.

Round 2

Reviewer 2 Report

Please, find the attached file.

Author Response

Thank you for your constructive comments. We have tried to address them.

We hope the new version would be mature enough for your acceptance for publication.

In detail:

Line 13: What is meant by multidrug resistant antibiotics? Thank you, for your indication. “multidrug resistant” was placed incorrectly in the sentence, authors have modified it to: “multidrug resistant superbugs”.

Line 41: Introduction must be focus on the problem the research dealt with and how will the authors solve this problem in addition to determining the gap in this point in previous researches. This comment is not taken into consideration: Thank you for your comment.

Authors added a new version of the introduction, specially modifying its last part. Current version remarks that there is a need for the discovery of new drugs against multidrug resistant bugs in the scientific community. Describes the current tendencies as the use on natural products from microbial fermentations, that there are evidences new molecules observed in the genomes of the microbial strains, and the techniques currently used to express those silent genomes towards the generation of libraries of extracts with potential new chemistry that are evaluated for antibacterial properties.

Finally, we have reflected in the introduction how the authors tried to contribute to the discovery of new antibacterial agents by surveying a high biodiversity source of actinomyces and exploiting the strains discovered by using current successful fermentation techniques, and how they found thanks to this approach, new antibiotic skeletons from the active extracts of the new Kribbella sp. CA-293567 strain.

Line 349: On which basis the authors selected the pathogens investigated. This comment was not considered: Thank you for your comment.

Authors indicated in lines 514 to 516 of the M&M Section the basis for the selection of the indicated pathogens panel that has been used: “Pathogen panel investigated was selected according to typical reference strains and procedures indicated by the NCSLI (National Conference of Standards Laboratories International)(Boulder, CO, USA).”

Line 525: The authors must select the selective media for the used pathogen. This comment was not considered: Thank you for your comment.

Selective media for the used pathogens, as well as all the details for the use of the pathogens in the panel for bioactivity characterization are described in the corresponding references for each assay [51 to 56]. It is also indicated in the different paragraphs of the 3.8 Antimicrobial Assays Section (Lines 508 to 540]. In detail, just for the information, the corresponding selective media were MRSA MB-5393: BHI (Brain Heart Infusion); A. baumannii MB-5973 and P. aeruginosa MB-5919: MHII (Muller Hinton Cation adjusted) and finally, A. fumigatus ATCC-46645 528 and C. albicans ATCC-64124: RPMI-1640 adjusted with MOPS to pH 7.0.

Line 541: Gram not gram. We have corrected all the ‘gram’s into ‘Gram’s. Thank you for detecting them.

Line 582: Where is the statistical analysis? Thank you for your comment.

Authors have added “as 10-point curves with 1:2 dilutions starting at 128 μg/mL in duplicate” to the 3.8 Antimicrobial Assays Section.

Regarding the statistical analysis for the robustness of the assay, it is performed within the indicated software from GendataTM (Screener). This statistical analysis is described in more detail in the assay methodology references [51 to 56]. For the information:

The MIC were defined as the lowest concentration of compound that inhibited ≥95% of the growth of a microorganism after overnight incubation. Absorbance at OD 612 nm was measured with an EnVision Microplate Reader (PerkinElmer) at T0 (zero time) and immediately after that, plates were statically incubated at 37 °C for 20 h. After this period, the assay plates were shaken using the DPC Micromix-5 and once more the absorbance at OD 612 nm was measured at Tf (final time). Percentage of growth inhibition was calculated using the following normalization:

%Inhibition= 100∙([(TfSample-T0Sample )-(TfBlank-T0Blank )])/([(TfGrowth-T0Growth)-(TfBlank-T0Blank)])

Where,

T0Sample = the absorbance of the strain growth in the presence of sample measured at zero time (before incubation)

TfSample = the absorbance of the strain growth in the presence of sample measured at final time (after incubation)

T0Growth= the absorbance of the strain growth in the absence of sample measured at zero time (before incubation)

TfGrowth = the absorbance of the strain growth in the absence of sample measured at final time (after incubation)

T0Blank = the absorbance of the broth medium (blank) measured at zero time (before incubation)

Reviewer 3 Report

All the comments have been addressed and the manuscript is revised accordingly. It is recommended to publish in the forthcoming issue of Molecules.

Author Response

Thank you very much for your very constructive comments and for accepting the proposed revised version of the manuscript